# Advances in Integrated Antimicrobial Resistance Surveillance and Control Strategies in Asia-Pacific Economic Cooperation Economies: Assessment of a Multiyear Building Capacity Project

**DOI:** 10.3390/antibiotics11081022

**Published:** 2022-07-29

**Authors:** Javiera Cornejo, Gabriela Asenjo, Sebastian Zavala, Lucas Venegas, Nicolás Galarce, Juan Carlos Hormazábal, Constanza Vergara-E, Lisette Lapierre

**Affiliations:** 1Departamento de Medicina Preventiva Animal, Facultad de Ciencias Veterinarias y Pecuarias, Universidad de Chile, Santiago 8820808, Chile; jacornej@uchile.cl (J.C.); gabiasenjo@gmail.com (G.A.); elias.szavalam@gmail.com (S.Z.); lucas.venegas@ug.uchile.cl (L.V.); nicolas.galarce@unab.cl (N.G.); 2Grupo Colaborativo Una Salud-Chile, Santiago 8820808, Chile; jchormazabal@ispch.cl (J.C.H.); constanza.vergara@achipia.gob.cl (C.V.-E.); 3Escuela de Medicina Veterinaria, Facultad de Ciencias de la Vida, Universidad Andrés Bello, Santiago 8370146, Chile; 4Subdepartamento de Enfermedades Infecciosas, Instituto de Salud Pública de Chile, Santiago 7780050, Chile; 5Agencia Chilena para la Inocuidad Alimentaria, Santiago 8320320, Chile

**Keywords:** antimicrobial resistance, surveillance, APEC, high economies, low–middle economies

## Abstract

Antimicrobial resistance (AMR) is a growing global health concern for both animal and public health, and collaborative strategies are needed to combat the threat. The level of awareness and funding for policies focused on reducing AMR varies between countries. The aim of this study was to compare the integrated surveillance systems for AMR in high and low–middle economies of the Asia-Pacific Economic Cooperation and determine whether there was any improvement from 2015 to 2018. We conducted a survey with a group of 21 countries at different development levels. Associations between the economic development level and the questions of AMR awareness and funding were established using Fisher’s exact test. Improvements were identified where countries established public policies for integrated surveillance of AMR. High economies showed greater advancement in several topics related to AMR than low–middle economies. The survey revealed that there is a better understanding surrounding the implications of the emergence of AMR in human medicine than in veterinary medicine, agriculture, and food production. Our results show that countries enhanced overall AMR surveillance over the 4-year-period; however, more research is needed concerning these advances, especially in low–middle economies and the food production sector.

## 1. Introduction

Antimicrobial resistance (AMR) is a global issue that threatens a century of progress in both animal and public health [1] and the sustainability of an effective global public health response to bacterial diseases [2], including antibiotic use for surgical procedures and chemotherapy [3]. It is widely accepted that the systematic use of antibiotics in human and veterinary medicine, including food production, puts every nation at risk. In 2019, it was estimated that globally 4.95 million deaths were due to AMR [4]. Therefore, this is an issue of utmost urgency [2], and it is essential to engage and unite all stakeholders around a shared vision and goals [1].

Given the increasing relevance of AMR, the Food and Agriculture Organization of the United Nations (FAO), the World Organization for Animal Health (WOAH), and the World Health Organization (WHO) Tripartite High-Level Coordination Forum set AMR as one of five priority issues and as a flagship topic for One Health, establishing an agreement to develop a joint action plan for AMR in 2014 [5]. In 2022, the United Nations Environment Programme joined the former Tripartite as an equal partner to form a new Quadripartite Collaboration for One Health in a signed memorandum of understanding [6]. The objectives of this group include coordination, promotion, and the development of a roadmap of the desired response to AMR in 2030 [7,8,9].

Using the official recommendations of WHO, FAO and WOAH, many high economies have established surveillance programs or systems to monitor AMR for antibiotic stewardship, while most low–middle economies have little to no programs to address this issue [10] due to limited resources, human capacity, laboratories, drugs, policies, and formal programs [11]. Consequently, some economies are working toward AMR suppression, while in others, antimicrobial consumption is increasing [12]. Some [13,14] have described that the development of AMR is dependent on the infections present in a particular country, economic level, and awareness of the population, pointing out that a lack of education could lead to overuse or misuse of antibiotics and that timely government policies could reduce this burden. Nevertheless and as reported by Cecchini et al. [15], less than 40% of countries have implemented control and prevention systems, and only 20% have national policies to reduce AMR. Furthermore, growth in animal production, the global food trade, and international travel allows resistant microorganisms to spread rapidly through humans, animals, and food [16]. However, the development of antimicrobial stewardship programs (ASPs) in some low–middle economies to improve antibiotic use in humans has resulted in increased compliance with antibiotic policy and reduced the duration of antibiotic treatment, cost of antibiotics, as well as antibiotic resistance, and healthcare-associated infections [17].

In 2015 and 2018, our research group surveyed two Asia-Pacific Economic Cooperation (APEC) projects, both focusing on AMR control strategies in humans and food production animals, to evaluate the status of AMR integrated surveillance in various APEC economies at different development levels [18]. The first survey in 2015 reported that only a few had implemented AMR control strategies throughout the food chain and that the human medical area had a better understanding of the implications of this phenomenon. Human medicine was also where most of the AMR control strategies in all economies were already implemented. The aim of this study was to compare the integrated surveillance systems and control strategies for AMR in high and low–middle economies of APEC and to determine whether there was any change between 2015 and 2018, especially in the animal health and food production sectors [15].

## 2. Results

A total of 14 (66.7%) APEC participant responses were received. The first group, called “High economies”, included Australia, Canada, the United States, Hong Kong, Japan, and New Zealand. The second group, named “Low–middle economies”, included Chile, Indonesia, Malaysia, Mexico, Peru, Philippines, Thailand, and Vietnam. These classifications resemble the Organization for Economic Co-operation and Development (OECD) classification terminology [19]. All responding economies answered the survey in full.

Table 1 shows the responses of all economies to all questions and their response for the previous period. Figure 1 plots the percentage of affirmative answers to eight questions about AMR surveillance in 2015 and 2018, disaggregated by economic level. Low–middle economies presented a similar or higher percentage of affirmative answers from 2015 to 2018, while high economies maintained their percentage of affirmative answers in six questions and increased their levels in two. Among high economies, 20% of respondents described AMR awareness as “High”, 60% as “Medium”, and 20% as “Low.” In low–middle economies, the values for the same levels of awareness were 13%, 38%, and 50%, respectively. The main AMR concerns for each respondent are shown in Table 2. The most stated concerns were “lack of alternatives to antibiotics” either in humans or food production animals and “use of antibiotics as growth promoters.”

The funding and reports of respondent surveillance systems can be observed in Table 3. The most common responses from high economies were “human AMR surveillance systems that receive funding” and “food chain AMR surveillance systems that receive funding”, while “human AMR surveillance systems that receive funding” and “national reports about the advances on human AMR updated in the last 5 years” were more important in low–middle economies. Additionally, of the 24 questions in the survey, 3 presented a significant *p*-value (≤0.05) (Table 4), representing a statistical association.

## 3. Discussion

Even when AMR in humans is recognized as a major global emergency problem [20], it is widely accepted that extensive use of antimicrobials in animal production systems is a major driver of multidrug resistance in bacteria [21]. Long-term subtherapeutic exposure to antibiotics can result in mutation enrichment and/or acquisition of mobile genetic elements that can confer a phenotype of increased resistance to these drugs [22]. The presence of antibiotic-resistant bacteria in people, animals, and food, regardless of their pathogenicity, constitutes a public health risk, as the genetic pool from which bacterial pathogens can acquire antibiotic resistance has increased in the environment [23]. For this reason, it is crucial that governments implement strategies encouraging the proper use of antibiotics, as well as controlling the emergence of antibiotic-resistant bacteria. Moreover, antimicrobial overuse or misuse is believed to be a major driver of AMR in south Asia. Therefore, surveillance programs might be an appropriate response to limit access to these drugs [4,24].

To better understand the awareness of different economies, regardless of the level of development, we conducted two APEC projects surveying participating economies. Our results show that since the first survey in 2015, there has been an improvement in all economies with respect to AMR surveillance in both human health and animal production. This could be attributed to the economies’ participation in the first APEC project and the subsequent development and improvement of surveillance systems. The least amount of improvement and greatest differences between high and low–middle economies were in the animal production sector. The shortfall occurs due to differences in resources, laboratories, legal frameworks, and human capacity. In low–middle economies, some areas of AMR research have been neglected [14,25,26].

The main concerns regarding AMR varied between high economies and low–middle economies. High economies emphasized the lack of alternatives to antibiotics and the lack of audits of veterinary farms and clinics. Low–middle economies highlighted the use of critically important antimicrobials in agroindustry and their lack of resources to act on this issue. Up-to-date knowledge on optimal antibiotic use was found to be higher in high economies for both human and veterinary use; meanwhile, in low–middle economies, it was only found for human antimicrobial use. This gap could be the result of limited animal health guidelines. In some cases, when these guidelines exist, they are not tailored to the end users or are only applicable to large-scale animal production. More research concerning the different characteristics of animal production in each economy is needed to develop customized guidelines that really address this problem.

All high economies surveyed received funding for both human and food chain surveillance, while only some low–middle economies received funding for human and food chain AMR surveillance. The same difference is observed in the proportion of reports about advances in surveillance in humans and in the food chain. Surveillance programs should integrate data originating from humans, food production animals, and retail meats [27]. AMR is present throughout the food chain, and different types of resistant bacteria have been found in different stages, including food production animals, animal products, and animal feed [28,29], emphasizing the need for integrated surveillance that considers humans, animals, and the environment. A One Health approach is needed to monitor AMR with reference to the WHO Global Action Plan.

Results indicate that by 2018, there was a better understanding of the implications of AMR occurrence in human medicine than in veterinary medicine, agriculture, and/or food production. Most economies have conducted AMR surveillance systems focused on human health, and only in the high economies have surveillance systems been developed for animal health. This is similar to that reported by Founou et al. [25], who pointed out that most low–middle economies have minimal or nonexistent programs or systems to monitor antibiotic use in nonhuman health settings (veterinary medicine, food, agriculture, or environment); thus, the true burden of antibiotic use is only partially documented and depends on where the studies are conducted.

It is known that AMR can only be controlled through a concerted global effort, led by heads of state and global institutions, and through coordinated action by the human and animal health sectors, agriculture, in collaboration with the food industry, and community organizations under the concept of One Health [27]. Our results show that there is no “one size fits all” solution for implementing effective AMR control strategies. Therefore, international multilateral organizations such as the Quadripartite must support countries to find tailored actions against AMR.

All economies have made progress since 2015 in the development and implementation of national action plans for AMR under a One Health approach. However, a new survey is needed since the COVID-19 emergency deprioritized planned activities and diverted human and financial resources. Furthermore, antimicrobial use in humans increased in 2020, fueled by COVID-19 prevention or treatment of bacterial coinfections [29]. All these factors may impact AMR surveillance actions, especially in veterinary or food production chains.

## 4. Materials and Methods

### 4.1. Survey Design

In 2018, a follow-up survey was sent to the 21 APEC economies previously surveyed by Lapierre et al. [18], including Australia, Brunei Darussalam, Canada, Chile, People’s Republic of China, Hong Kong, Indonesia, Japan, Republic of Korea, Malaysia, Mexico, New Zealand, Papua New Guinea, Peru, the Philippines, Russia, Singapore, Chinese Taipei, Thailand, the United States, and Vietnam. This survey, the same from 2015, was also based on the questionnaire provided by the Pan American Health Organization (PAHO) “Rapid Assessment Tool for the Analysis of the Country’s Situation on Antimicrobial Resistance” [30], a multi-stage analytical tool to assess the situation at the national level. This tool was based on existing WHO assessment tools and reflected the elements contained in the policy package to address AMR that was issued on World Health Day 2011 [31]. The survey was made up of 24 questions, including open questions, Likert scale questions from 1 to 4 (High, Medium, Low, None) plus the “Not known” option, closed questions with “Yes”, “No”, and “Not known” options and closed questions with specific options. Additionally, the survey was divided into four topics: (i) general information about the use of antimicrobials and the actions taken to control it, (ii) existence, permanence, and funding of surveillance systems in humans and animals, (iii) legal and regulatory aspects concerning antimicrobials and AMR surveillance systems, and (iv) integration of human health, animal health, and food production (full survey in Appendix A). The survey was validated by a group of five experts in the field belonging to national and international governmental agencies and multilateral organizations, who were asked to assess the coherence, clarity, sufficiency, and relevance of the survey [32]. Once validated, the survey was sent through the APEC secretariat to the focal points. It was requested that respondents were from the government and were policy decision-makers concerning AMR, antibiotic use, and registration for human and food production animals. Groups were given one month to respond.

### 4.2. Economy Development Level

APEC economies are very different and have different degrees of development. Therefore, they were divided into two groups, “High economies” and “Low–middle economies”, according to the classification of APEC economies considered “travel-eligible” [19]. This APEC classification considers parameters used to establish the travel-eligible economies based on the economic development and income status of APEC members in 2002, which were decided and approved by consensus. Differences in AMR surveillance levels and control strategies between these two groups were compared to analyze change over time.

### 4.3. Statistical Analysis

Multiple analyses were performed for each question with the response data “Yes”, “No”, and “Not known” and the economic level category “High”, “Medium”, “Low”, “None”, and “Not known” using Fisher’s exact test [33]. Only the questions that were repeated in both surveys were used to compare the differences between 2015 and 2018. All statistical analyses were performed in R 4.0.2 (Integrated Development for R. RStudio, Inc., Boston, MA, USA) [34].

## 5. Conclusions

This pilot survey and its results from different years of reporting contribute to the global objective of tackling AMR in both high and low–middle economies. Cooperation between economies is a key principle in decreasing the gaps between groups and ensuring every economy can implement a viable national plan to control AMR. This model could be expanded to other countries or international entities to encourage the flow of information about AMR surveillance and control strategies. It is very important that high economies, which have surveillance systems in place (i.e., US NARMS, CIPARS-Canada), can share their knowledge and experiences with low–middle economies, making implementation more effective and efficient since they can communicate not only good results obtained but also their failures. AMR must be looked at as a global public threat, so every country has a shared responsibility and should collaborate to try to decrease AMR and help prioritize areas where these actions are most critical, such as food production surveillance and its integration with human AMR surveillance. Finally, it is necessary for a third survey assessment in the APEC economies to measure the impact of their advances due to the health, economic, and social effects of COVID-19.

## Figures and Tables

**Figure 1 antibiotics-11-01022-f001:**
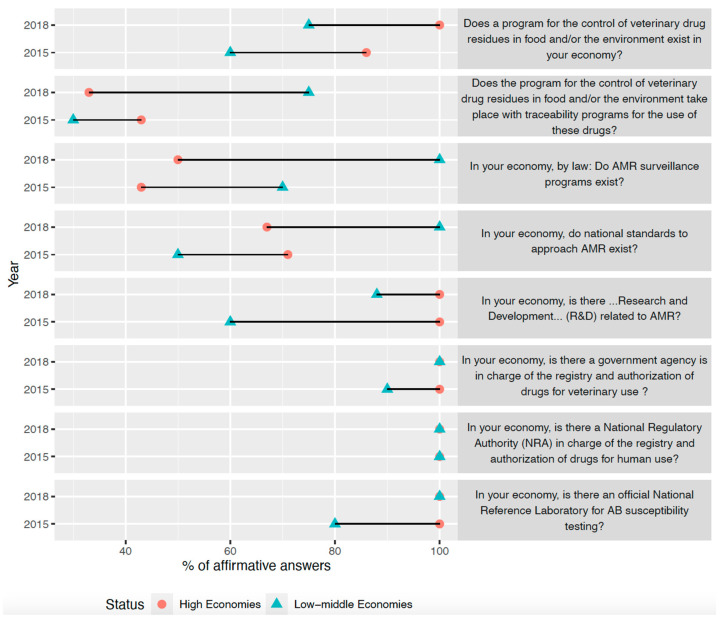
Levels of antimicrobial surveillance in the Asia-Pacific region among high and low–middle economies.

**Table 1 antibiotics-11-01022-t001:** Responses concerning the existence of antimicrobial surveillance in the Asia-Pacific Region among high and low–middle economies: NK, not known. Q1: Does a program for the control of veterinary drug residues in food and/or the environment exist in your economy? Q2: Does the program for the control of veterinary drug residues in food and/or the environment take place with traceability programs for the use of these drugs? Q3: In your economy, by law, do AMR surveillance programs exist? Q4: In your economy, do national standards to approach AMR exist? Q5: In your economy, is there “research and development” (R&D) related to AMR? Q6: In your economy, is there a government agency in charge of the registry and authorization of drugs for veterinary use? Q7: In your economy, is there a national regulatory authority (NRA) in charge of the registry and authorization of drugs for human use? Q8: In your economy, is there an official national reference laboratory for AB susceptibility testing?

Country	Question 1		Question 2		Question 3		Question 4		Question 5		Question 6		Question 7		Question 8	
Year	2015	2018	2015	2018	2015	2018	2015	2018	2015	2018	2015	2018	2015	2018	2015	2018
Australia	YES	YES	YES	NO	NO	NO	YES	YES	YES	YES	YES	YES	YES	YES	YES	YES
Canada	YES	YES	NO	YES	YES	YES	NO	NO	YES	YES	YES	YES	YES	YES	YES	YES
United States	YES	YES	NO	NO	NO	YES	NK	YES	YES	YES	YES	YES	YES	YES	YES	YES
Hong Kong	YES	YES	NO	NK	NO	NK	YES	NO	YES	YES	YES	YES	YES	YES	YES	YES
Japan	NK	YES	NO	NO	YES	YES	YES	YES	YES	YES	YES	YES	YES	YES	YES	YES
New Zealand	YES	YES	YES	YES	NO	NO	YES	YES	YES	YES	YES	YES	YES	YES	YES	YES
Taipei	YES	---	YES	---	YES	---	YES	---	YES	---	YES	---	YES	---	YES	---
TOTAL (%)	85.7	100	42.9	33.3	42.9	50	71.4	66.7	100	100	100	100	100	100	100	100
Chile	YES	YES	YES	YES	YES	YES	YES	YES	NO	YES	YES	YES	YES	YES	YES	YES
China	NK	---	NK	---	YES	---	NK	---	YES	---	YES	---	YES	---	NK	---
Philippines	YES	YES	NO	YES	YES	YES	YES	YES	YES	YES	YES	YES	YES	YES	YES	YES
Indonesia	NO	YES	NO	NK	YES	YES	NO	YES	NK	YES	YES	YES	YES	YES	YES	YES
Malaysia	YES	YES	YES	YES	YES	YES	YES	YES	YES	YES	YES	YES	YES	YES	YES	YES
Mexico	YES	YES	YES	YES	YES	YES	YES	YES	YES	NO	YES	YES	YES	YES	YES	YES
Papua New Guinea	NK	---	NO	---	NO	---	NO	---	NK	---	NK	---	YES	---	YES	---
Peru	YES	YES	NO	YES	NO	YES	YES	YES	YES	YES	YES	YES	YES	YES	YES	YES
Thailand	YES	NO	NK	NK	NK	YES	NK	YES	YES	YES	YES	YES	YES	YES	YES	YES
Vietnam	NK	NO	NK	YES	YES	YES	NO	YES	NK	YES	YES	YES	YES	YES	NK	YES
TOTAL (%)	60	75	30	75	70	100	50	100	60	87.5	90	100	100	100	80	100

**Table 2 antibiotics-11-01022-t002:** Main concerns for addressing AMR in high and low–middle economies.

Question: What Is the Most Common Concern in Addressing the Issue of AMR in Your Economy?	High Economies (%)	Low–Middle Economies (%)
Lack of antibiotic (AB) registry	0	0
Lack of alternatives to AB	83	50
Use of AB as growth promoters	17	63
Lack of legislation to monitor or control the use of AB	17	50
Lack of audit of AB use at farm or veterinary clinics	83	50
Nonexistence of AMR surveillance programs in human health	0	13
Nonexistence of AMR surveillance programs in food/animals	17	38
Use of critically important AB for humans used in agroindustry	17	88
Lack of prescription for veterinary AB sales	0	25
Lack of technical knowledge of AMR	17	38
Lack of resources to take action on AMR	33	88

**Table 3 antibiotics-11-01022-t003:** Surveillance funding and reports in high and low–middle economies.

Question	High Economies (%)	Low–Middle Economies (%)
Do human AMR surveillance systems receive funding?	100	75
Do food chain AMR surveillance systems receive funding?	100	50
Have the national reports about advances in human AMR been updated in the last 5 years?	83	75
Have the national reports about advances in food chain AMR been updated in the last 5 years?	50	25

**Table 4 antibiotics-11-01022-t004:** Questions that show associations between economic level and positive response.

Question	*p*-Value
In your economy, by law, does an AMR surveillance program exist?	0.002997
Of the following matters, which ones are of most concern in addressing the issue of AMR in your economy? Response: “Use of critically important antibiotics for humans used in agroindustry.”	0.02564
Does the control of veterinary drugs take place with traceability programs?	0.04545

## Data Availability

Not applicable.

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
