# Peer review of "Advances in Integrated Antimicrobial Resistance Surveillance and Control Strategies in Asia-Pacific Economic Cooperation Economies: Assessment of a Multiyear Building Capacity Project"

_antibiotics, 2022, doi:10.3390/antibiotics11081022_

Round 1

Reviewer 1 Report

This manuscript compared the government systems/policies of controlling and monitoring AMR status and development in 21 countries in 2015 and 2018 and showed these countries have some improvement on AMR surveillance.

The strength of this manuscript is that it presented the globe change trend on AMR surveillance and the rising public notice on AMR issues. The key weakness is that it is sort of the extension of the 2015 study (reference 11).Although it is still meaningful but lack of novelty.

The manuscript is well written. However, to make it more informative, it would be good to add a table about the AMR surveillance programs' or systems' name. For instance, I know there is a AMR surveillance system in US, but not sure if the one I know is the one they mentioned in the manuscript. This will also increase the citation for this manuscript. So I would strongly recommend authors to add this. Another weakness is that authors mentioned about the changes in "a 4-years-period". So I was expecting to see the data from 2015, 2016, 2017 and 2018. But I only see 2015 and 2018. How can this count as "a 4-years-period"? It should be changes after 4 years. The last minor correction is to delete the "-" in Table 1's title line "Emerging mar-kets".

The manuscript is well written and easy to follow. The only suggestion would be for Table 1. It can be re-organized for better presence. The current format made it very difficult for audience to read.

Overall, I think this manuscript is OK for publish as its current status, but would have more scientific significances if the 21 countries' AMR policy/system names can be added.

Author Response

Reviewer 1:

This manuscript compared the government systems/policies of controlling and monitoring AMR status and development in 21 countries in 2015 and 2018 and showed these countries have some improvement on AMR surveillance.
The strength of this manuscript is that it presented the globe change trend on AMR surveillance and the rising public notice on AMR issues. The key weakness is that it is sort of the extension of the 2015 study (reference 11). Although it is still meaningful but lack of novelty.

1) The manuscript is well written. However, to make it more informative, it would be good to add a table about the AMR surveillance programs' or systems' name. For instance, I know there is a AMR surveillance system in US, but not sure if the one I know is the one they mentioned in the manuscript. This will also increase the citation for this manuscript. So I would strongly recommend authors to add this.

R: We are well aware that some of the surveyed economies have established AMR surveillance programs (e.g. CIPARS-Canada and NARMS-USA), but other countries have failed to establish them. For this reason, and in order to avoid bias in the information, we decided to mention only a few in the text.

2) Another weakness is that authors mentioned about the changes in "a 4-years-period". So I was expecting to see the data from 2015, 2016, 2017 and 2018. But I only see 2015 and 2018. How can this count as "a 4-years-period"? It should be changes after 4 years.

R: We understand that the phrase "in a 4-year-period" may have led to confusion about the results. For this reason, and to be explicit that the results were obtained only in the last year, we added "over the 4-year-period" when necessary.

3) The last minor correction is to delete the "-" in Table 1's title line "Emerging mar-kets".

R: We corrected the title of Table 1.

4) The manuscript is well written and easy to follow. The only suggestion would be for Table 1. It can be re-organized for better presence. The current format made it very difficult for audience to read.

R: We agree. According to this reviewer’s suggestion and in order to make it easier for readers to understand, we have changed the table format.

5) Overall, I think this manuscript is OK for publish as its current status, but would have more scientific significances if the 21 countries' AMR policy/system names can be added.

R: As explained above, decided to mention only a few of those programs in the text.

Reviewer 2 Report

According to the authors, this study is a follow-up study from 2017, published in Asia Pac J Public Health. The study is based on a survey among different countries concerning AMR strategies. The results of this study are based on only 14 answers, which is, in my opinion, much too low to show an accurate picture of this situation on this matter. Besides, the study aims to compare advanced and emerging markets, but none of the countries from Europe has been included in the survey. There is also not enough data on the methodology of analyzing data. 

Author Response

Reviewer 2:

1) According to the authors, this study is a follow-up study from 2017, published in Asia Pac J Public Health. The study is based on a survey among different countries concerning AMR strategies. The results of this study are based on only 14 answers, which is, in my opinion, much too low to show an accurate picture of this situation on this matter. Besides, the study aims to compare advanced and emerging markets, but none of the countries from Europe has been included in the survey. There is also not enough data on the methodology of analyzing data.

R: The aim of this study was to compare the integrated surveillance systems for AMR in advanced economies and emerging markets of the Asia-Pacific Economic Cooperation (APEC) and to determine if there was any change between 2015 and 2018. The APEC is the premier Asia-Pacific economic forum, whose primary goal is to support sustainable economic growth and prosperity in the Asia-Pacific region. Currently, APEC has 21-member countries, of which 14 responded to our survey, representing more than half of its members (66.7%). Only APEC countries were considered for this study.

Reviewer 3 Report

Thank you for this paper. However - I have a number of concerns as current key activities, etc., including a number of those co-ordinated by the WHO were not mentioned. I also have concerns with the Methodology as it currently documented. Addressing these will add to the utility of the paper/ its generalisability across countries

Key issues include:

A) Introduction

i) Lines 34 - 41 - I agree that the WHO Global Action Plan on AMR is a very important document that led to the development of National Action Plans across countries with discussions on surveillance, activities, etc., to reduce AMR. I cannot see any reference to any NAP from the countries selected in the paper/ any difference between developed/ developing countries - this is important going forward. There have been publications on this, e.g. Iwu CD et al. An insight into the implementation of the global action plan on antimicrobial resistance in the WHO African region: A roadmap for action. Int J Antimicrob Agents. 2021;58:106411; Chua AQ et al. An analysis of national action plans on antimicrobial resistance in Southeast Asia using a governance framework approach. Lancet Reg Health West Pac. 2021;7:100084 and Harant A. Assessing transparency and accountability of national action plans on antimicrobial resistance in 15 African countries. Antimicrob Resist Infect Control. 2022;11:15 to name a few alongside World Health Organization (WHO), Food and Agriculture Organization of the United Nations (FAO) and World Organisation for Animal Health (OIE). MONITORING GLOBAL PROGRESS ON ANTIMICROBIAL RESISTANCE: TRIPARTITE AMR COUNTRY SELF-ASSESSMENT SURVEY (TRACSS) 2019-2020 GLOBAL ANALYSIS REPORT. Available at URL: https://www.who.int/publications/i/item/monitoring-global-progress-on-antimicrobial-resistance-tripartite-amr-country-self-assessment-survey-(tracss)-2019-2020 - FAO, OIE, WHO. Global Database for the Tripartite Antimicrobial Resistance (AMR) Country Self-assessment Survey (TrACSS). 2020 - 2021. Available at URL: http://amrcountryprogress.org/#/map-view - with the WHO recently issuing an update handbook to assist countries - WHO implementation handbook for national action plans on antimicrobial resistance: guidance for the human health sector - Available at URL: https://www.who.int/publications/i/item/9789240041981  - these build on lines 42 - 47 (not mentioned). This is because we know that resources (personnel and funding) can be an issue especially in developing countries to fully implement NAPs, e.g. Saleem Z, Godman B, Azhar F, Kalungia AC, Fadare J, Opanga S, et al. Progress on the national action plan of Pakistan on antimicrobial resistance (AMR): a narrative review and the implications. Expert Review of Anti-infective Therapy. 2022;20:71-93. Recent references discussing the extent of AMR among different Regions as well as the financial impact include Global burden of bacterial antimicrobial resistance in 2019: a systematic analysis. Lancet. 2022 and Hofer U. The cost of antimicrobial resistance. Nat Rev Microbiol. 2019;17:3.    

ii) Lines 49 - 61 - some of these are old references including Ref 9 (URL Not recorded is https://www.oecd.org/els/health-systems/Antimicrobial-Resistance-in-G7-Countries-and-Beyond.pdf) - need to build on this with some of the above references. Developing and implementing NAPs can be challenging in LMICs - we see this for e.g. ASPs - Cox JA et al. Antibiotic stewardship in low- and middle-income countries: the same but different? Clin Microbiol Infect. 2017;23:812-8 - however this is changing as seen with e.g. Akpan et al. Implementation of antimicrobial stewardship programmes in African countries: a systematic literature review. J Glob Antimicrob Resist. 2020;22:317-24.

iii) Lines 67 - 69 - why stop at 2018 with a number of recent developments and initiatives including instigation and monitoring of National Action Plans on AMR in a number of countries. This makes no sense without a robust explanation

iv) Need to state that the paper is more about animal use/ food production than the use/ misuse of antimicrobials in humans - and why this is the case - when AMR in humans is increasingly seen as the next pandemic in view of increasing morbidity and mortality rates

B) Results - Lines 71 - 76 (and in the Methodology) - why were these emerging countries chosen over multiple others? If say one group, e.g. low-middle, high-middle, etc. - then need to convey this in the title as 'emerging' can refer to low-income countries as well moving up. If a tight group - this needs to be reflected in the Title as well - together with a good explanation in the Methodology as to why this particular group was chosen over the others (do not see this currently). In addition - need to reflect in the Title that only APEC countries with no mention at all of e.g. Sub-Saharan African countries who have the greatest burden of infectious diseases world wide and also the greatest burden of AMR (Global burden of bacterial antimicrobial resistance in 2019: a systematic analysis. Lancet. 2022)  

C) Discussion

i) Lines 111 - 120 really need to go into the Introduction to help set the scene for the study - this alongside as stated above why most discussion on animal health/ food production and not on programmes to ensure appropriate use of antibiotics in humans (equal part) especially with recent WHO initiatives in this area alongside the growth/ monitoring of National Action Plans on AMR (above references)

ii) Ref 17 lines 157 - 160 is now old. I would like to see more Discussion on the way forward in light of Global WHO activities including WHO. Global Antimicrobial Resistance and Use Surveillance System (GLASS) Report: 2021 -  https://www.who.int/publications/i/item/9789240027336 to add weight/ utility to the paper

D) Methodology

i) As state earlier - why not beyond 2018

ii) Lines 163 - 167 - who was the survey sent to in these various countries - no mention of the personnel, their position, etc., to add robustness to any findings. This needs to be rectified

iii) Lines 180 - 183 - Please give more details of the validation process - this is very thin currently

E) Conclusion

These are very thin and general at the moment - what do you suggest in light of WHO and other activities with your knowledge - goo to expand on this

Author Response

Reviewer 3:

Thank you for this paper. However - I have a number of concerns as current key activities, etc., including a number of those coordinated by the WHO were not mentioned. I also have concerns with the Methodology as it currently documented. Addressing these will add to the utility of the paper/ its generalisability across countries

Key issues include:

  1. A) Introduction

1) Lines 34 - 41 - I agree that the WHO Global Action Plan on AMR is a very important document that led to the development of National Action Plans across countries with discussions on surveillance, activities, etc., to reduce AMR. I cannot see any reference to any NAP from the countries selected in the paper/ any difference between developed/ developing countries - this is important going forward. There have been publications on this, e.g. Iwu CD et al. An insight into the implementation of the global action plan on antimicrobial resistance in the WHO African region: A roadmap for action. Int J Antimicrob Agents. 2021;58:106411; Chua AQ et al. An analysis of national action plans on antimicrobial resistance in Southeast Asia using a governance framework approach. Lancet Reg Health West Pac. 2021;7:100084 and Harant A. Assessing transparency and accountability of national action plans on antimicrobial resistance in 15 African countries. Antimicrob Resist Infect Control. 2022;11:15 to name a few alongside World Health Organization (WHO), Food and Agriculture Organization of the United Nations (FAO) and World Organisation for Animal Health (OIE). MONITORING GLOBAL PROGRESS ON ANTIMICROBIAL RESISTANCE: TRIPARTITE AMR COUNTRY SELF-ASSESSMENT SURVEY (TRACSS) 2019-2020 GLOBAL ANALYSIS REPORT. Available at URL: https://www.who.int/publications/i/item/monitoring-global-progress-on-antimicrobial-resistance-tripartite-amr-country-self-assessment-survey-(tracss)-2019-2020 - FAO, OIE, WHO. Global Database for the Tripartite Antimicrobial Resistance (AMR) Country Self-assessment Survey (TrACSS). 2020 - 2021. Available at URL: http://amrcountryprogress.org/#/map-view - with the WHO recently issuing an update handbook to assist countries - WHO implementation handbook for national action plans on antimicrobial resistance: guidance for the human health sector - Available at URL: https://www.who.int/publications/i/item/9789240041981  - these build on lines 42 - 47 (not mentioned). This is because we know that resources (personnel and funding) can be an issue especially in developing countries to fully implement NAPs, e.g. Saleem Z, Godman B, Azhar F, Kalungia AC, Fadare J, Opanga S, et al. Progress on the national action plan of Pakistan on antimicrobial resistance (AMR): a narrative review and the implications. Expert Review of Anti-infective Therapy. 2022;20:71-93. Recent references discussing the extent of AMR among different Regions as well as the financial impact include Global burden of bacterial antimicrobial resistance in 2019: a systematic analysis. Lancet. 2022 and Hofer U. The cost of antimicrobial resistance. Nat Rev Microbiol. 2019;17:3.

R: Thank you for all these recommendations. In order to improve the article and to use references that are as up to date as possible, we decided to include several of these references in the introduction and discussion.

  1. ii) Lines 49 - 61 - some of these are old references including Ref 9 (URL Not recorded is https://www.oecd.org/els/health-systems/Antimicrobial-Resistance-in-G7-Countries-and-Beyond.pdf) - need to build on this with some of the above references. Developing and implementing NAPs can be challenging in LMICs - we see this for e.g. ASPs - Cox JA et al. Antibiotic stewardship in low- and middle-income countries: the same but different? Clin Microbiol Infect. 2017;23:812-8 - however this is changing as seen with e.g. Akpan et al. Implementation of antimicrobial stewardship programmes in African countries: a systematic literature review. J Glob Antimicrob Resist. 2020;22:317-24.

R: Thank you for the recommended references, which were added to the article. We copied and pasted the URL you mentioned and were able to open the pdf document we cited. In any case, we pasted the URL again in the references in case some misspelling was made in the first attempt.

iii) Lines 67 - 69 - why stop at 2018 with a number of recent developments and initiatives including instigation and monitoring of National Action Plans on AMR in a number of countries. This makes no sense without a robust explanation.

R: The two projects that we compared were developed under APEC SCSC Food Safety Cooperation Forum priorities and funding, after 2018 there have been no more projects in the same scope in the APEC Forum. However, we improved the language in the paragraph for better understanding.

  1. iv) Need to state that the paper is more about animal use/ food production than the use/ misuse of antimicrobials in humans - and why this is the case - when AMR in humans is increasingly seen as the next pandemic in view of increasing morbidity and mortality rates.

R: We agree. According to this comment, we emphasize this in the text.

  1. B) Results - Lines 71 - 76 (and in the Methodology) - why were these emerging countries chosen over multiple others? If say one group, e.g. low-middle, high-middle, etc. - then need to convey this in the title as 'emerging' can refer to low-income countries as well moving up. If a tight group - this needs to be reflected in the Title as well - together with a good explanation in the Methodology as to why this particular group was chosen over the others (do not see this currently). In addition - need to reflect in the Title that only APEC countries with no mention at all of e.g. Sub-Saharan African countries who have the greatest burden of infectious diseases world wide and also the greatest burden of AMR (Global burden of bacterial antimicrobial resistance in 2019: a systematic analysis. Lancet. 2022) .

R: Thank you for your recommendation, APEC economies was included in the tittle, a reference to APEC food safety cooperation forum funding projects were added to the methodology and the economies are now referred to as High economies and Low-middle economies to resemble the High income and LMIC´s nomenclature used by OECD.

  1. C) Discussioni) Lines 111 - 120 really need to go into the Introduction to help set the scene for the study - this alongside as stated above why most discussion on animal health/ food production and not on programmes to ensure appropriate use of antibiotics in humans (equal part) especially with recent WHO initiatives in this area alongside the growth/ monitoring of National Action Plans on AMR (above references).

R: Thank you for the recommendation, to address your concern we added clarifying text in the introduction explaining why the results showed in this article are more focused on the food production animal/veterinary sector.

  1. ii) Ref 17 lines 157 - 160 is now old. I would like to see more Discussion on the way forward in light of Global WHO activities including WHO. Global Antimicrobial Resistance and Use Surveillance System (GLASS) Report: 2021 -  https://www.who.int/publications/i/item/9789240027336 to add weight/ utility to the paperD) Methodologyi)

As state earlier - why not beyond 2018.

R: Same answer to your earlier statement.

  1. ii) Lines 163 - 167 - who was the survey sent to in these various countries - no mention of the personnel, their position, etc., to add robustness to any findings. This needs to be rectified.

R: We agree. In this case, text clarifying this issue was added.

iii) Lines 180 - 183 - Please give more details of the validation process - this is very thin currently.

R: According to reviewer’s suggestion, the reference on the validation process was added for clarification.

  1. E) Conclusion

These are very thin and general at the moment - what do you suggest in light of WHO and other activities with your knowledge - goo to expand on this.

R: We added to the conclusion in line with the discussion. Thank you for the recommendation.

Round 2

Reviewer 2 Report

The manuscript has been corrected accordingly. 

Author Response

Thank you very much for your review.